# Intracellular presence of *Helicobacter pylori* antigen and genes within gastric and vaginal *Candida*

Tingxiu Yang[1,2,3,4☉], Jia Li[1,5☉], Yuanyuan Zhang[1,6‡], Zhaohui Deng[1‡], Guzhen Cui[1,3‡], Jun Yuan[3‡], Jianchao Sun[1‡], Xiaojuan Wu[1,4‡], Dengxiong Hua[1‡], Song Xiang[1‡], Zhenghong Chen[1,3,4]*

1 Key Laboratory of Microbiology and Parasitology of Education Department of Guizhou, School of Basic Medical Science/Joint Laboratory of Helicobacter Pylori and Intestinal Microecology of Affiliated Hospital of Guizhou Medical University, Guizhou Medical University, Guiyang, China, 2 Department of Hospital Infection and Management, Guizhou Provincial People's Hospital, Guiyang, China, 3 Key Laboratory of Endemic and Ethnic Diseases of Ministry of Education/Key Laboratory of Environmental Pollution Monitoring and Disease Control, Ministry of Education, Guizhou Medical University, Guiyang, China, 4 Scientific Research Center, School of Basic Medical Science, Guizhou Medical University Guiyang, Guiyang, China, 5 Department of Clinical Laboratory, Jinyang Affiliated Hospital of Guizhou Medical University, Guiyang, China, 6 Department of Gastroenterology, People's Hospital of Qiannan Prefecture, Guizhou, China

☉ These authors contributed equally to this work.
‡ These authors also contributed equally to this work.
* chenzhenghong@gmc.edu.cn

**Data Availability Statement:** All relevant data are within the manuscript and its Supporting Information files.

## Abstract

### Background

*Helicobacter pylori* infections are generally acquired during childhood and affect half of the global population, but its transmission route remains unclear. It is reported that *H. pylori* can be internalized into *Candida*, but more evidence is needed for the internalization of *H. pylori* in human gastrointestinal *Candida* and vaginal *Candida*.

### Methods

*Candida* was isolated from vaginal discharge and gastric mucosa biopsies. We PCR-amplified and sequenced *H. pylori*-specific genes from *Candida* genomic DNA. Using optical and immunofluorescence microscopy, we identified and observed bacteria-like bodies (BLBs) in *Candida* isolates and subcultures. Intracellular *H. pylori* antigen were detected by immunofluorescence using Fluorescein isothiocyanate (FITC)-labeled anti-*H. pylori* IgG antibodies. Urease activity in *H. pylori* internalized by *Candida* was detected by inoculating with urea-based Sabouraud dextrose agar, which changed the agar color from yellow to pink, indicating urease activity.

### Results

A total of 59 vaginal *Candida* and two gastric *Candida* strains were isolated from vaginal discharge and gastric mucosa. Twenty-three isolates were positive for *H. pylori* 16S rDNA, 12 were positive for *cagA* and 21 were positive for *ureA*. The BLBs could be observed in

**Funding:** This work was supported by grants from the National Natural Science Foundation of China (No. 81860353), the Basic Research Program of Guizhou Science and Technology Plan (No. ZK [2022] 341), the Science and Technology Fund Project of Guizhou Health Commission (No. gzwkj2022-521) and the Scientific Foundation of Guizhou Medical University (No. 20NSP005), and the Foundation of Key Laboratory of Microbiology and Parasitology of Education Department, Guizhou (No. QJJ [2022] 019). These funding obtained from Zhenghong Chen, who had a role in the study design, data collection and analysis, decision to publish, or preparation of the manuscript.

**Competing interests:** All authors declare no conflicts of interest.

*Candida* cells, which were positive for *H. pylori* 16S rDNA, and were viable determined by the LIVE/DEAD BacLight Bacterial Viability kit. Fluorescein isothiocyanate (FITC)-conjugated antibodies could be reacted specifically with *H. pylori* antigen inside *Candida* cells by immunofluorescence. Finally, *H. pylori*-positive *Candida* remained positive for *H. pylori* 16S rDNA even after ten subcultures. Urease activity of *H. pylori* internalized by *Candida* was positive.

## Conclusion

In the form of BLBs, *H. pylori* can internalize into gastric *Candida* and even vaginal *Candida*, which might have great significance in its transmission and pathogenicity.

## Introduction

Discovered in 1983, *Helicobacter pylori* (originally *Campylobacter pylori*) is a spiral-shaped, gram-negative bacterium that often causes acute and chronic gastritis and peptic ulcer disease [1]. It is also closely associated with the development of gastric mucosa-associated lymphoid tissue lymphoma and gastric cancer [1]. In 1994, the International Agency for Research on Cancer of the World Health Organization classified *H. pylori* as a class I carcinogen [2]. *H. pylori* infection is a public health problem worldwide [3], and approximately half of the world's population is infected with it [4]. *H. pylori* infection is especially higher in less developed countries such as Jordan, where 88.6% of people aged between 15 and 81 years are infected [5]. The infection is generally acquired during childhood and can remain asymptomatic, but *H. pylori* often exhibits a long-term and chronic infection process [6–8]. The child infection rates are typically around 30%: in Cairo, 33% of children under age of 6 years are infected [9]; in Poland, 32.01% of children between the ages of 6 months and 18 years are infected [10]; and in Changsha, China, 30.6% of infants and toddlers are infected [11].

The main transmission routes for *H. pylori* are thought to be fecal-oral [12] and oral-oral [13,14], and these routes occur through ingesting contaminated food or water [12,14], communal meals, sharing towels, receiving pre-chewed food from the mother and physical acts such as kissing [15,16]. From this discovery, questions arise about whether close contact, such as communal meals, sharing towels, or kissing can lead to the spread of *H. pylori* infection [17].

Some studies have suggested that mothers with *H. pylori* infection may be the main transmission source during childhood [8,18]. A subsequent study confirmed the mother-to-child transmission of *H. pylori*, a phenomenon that can be attributed to mothers being the primary caregivers and children having more opportunities for oral-oral and fecal-oral transmission [19]. However, *H. pylori* is extremely fragile and dies quickly after exposure to the air, and in our experiments, we failed to isolate viable isolates from saliva and feces of 50 patients infected with *H. pylori* (unpublished), most studies reporting on detection of *H. pylori* in saliva or dental plaque have been performed by polymerase chain reaction (PCR) methods [20–22]. Hamada et al. did not culture *H. pylori* from the oral cavity or fecal samples, and the *ureA* gene of *H. pylori* was positive by PCR amplification [23]. Mao *et al.* presented a critical discussion of previous studies investigating the potential colonization of the oral cavity by *H. Pylori* [17]. Therefore, the oral cavity is not a suitable host for *H. pylori* colonization.

As early as 2013, researchers have proposed that vaginal yeast is the primary reservoir of *H. pylori*, with the bacteria transmitted from *H. pylori*-positive mothers to newborns during

vaginal delivery or contact with the hospital environment and healthcare workers [24]. However, there has been little focus on investigating alternate transmission routes for *H. pylori* beyond the commonly recognized oral-oral or fecal-oral routes. Therefore, this study aimed to determine whether *H. pylori* is present in *Candida* isolated from vaginal discharge of pregnant women, as well as vaginal discharge and gastric mucosa of patients with digestive diseases.

In this study, *Candida* was isolated from vaginal discharge in pregnant women and female patients with gastric disease and from gastric mucosa biopsies in the female patients. Using optical and fluorescence microscopy, we identified and observed the bacteria-like bodies (BLBs) in isolated *Candida* and subcultures. We then PCR-amplified and sequenced *H. pylori* specific 16S rDNA and *cagA* genes from *Candida* cells. Finally, we separately diagnosed *H. pylori* infection in pregnant women and female patients using the *H. pylori* antibody test and urea breath test (UBT). The aim of this study was to detect intracellular *H. pylori* in vaginal and gastric *Candida* isolates and to explore the potential for transmission of *H. pylori* from mother to newborn through vaginal *Candida* during delivery.

## Material and methods

### Ethics approval and consent to participate

This study was approved by the Human Medical Ethics Committee of Guizhou Medical University in April 2021, following the Ethics Review Document No. 141. All procedures complied with the Declaration of Helsinki. Verbal informed consent was obtained from all participants for their anonymized information to be published in this article.

### Samples

This study included 50 pregnant women who underwent prenatal examination in the obstetrics department of Jinyang Affiliated Hospital at Guizhou Medical University from May to October 2021. Samples of vaginal discharge and serum were collected. In addition, 27 gastric biopsies and 22 vaginal discharges were collected from 42 female patients with gastric disease at the Department of Gastroenterology in the People's Hospital of Qiannan Prefecture (Guizhou Province, China) from May to October 2021. Seven female patients provided samples of both gastric mucosa and vaginal discharge. However, only vaginal discharge samples were obtained from another 15 female patients who did not undergo gastroscopic examination. Finally, 20 individuals were outpatients; therefore, only gastric mucosa samples were collected.

Infection in pregnant women was diagnosed via a quantitative detection of serum *H. pylori* antibodies. An *H. pylori* antibody test kit (WAN TAI BRD, Beijing, China) for performing latex immunoturbidimetry (ARCHITECT c16000, Abbott, USA) was used to determine *H. pylori* IgG. Finally, *H. pylori* infection in female patients with gastric disease was diagnosed using the $^{14}$C-urea breath test (UBT).

### *Candida* isolation and identification

Samples of vaginal discharge and homogenized gastric mucosa were inoculated directly on CHROMagar *Candida* medium (CHROMagar, Paris, France) following the manufacturer's protocol and then incubated at 37˚C for 24–48 h under aerobic conditions. *Candida* species were identified based on the color characteristics of different colonies, such as green-blue (*C. albicans*), metallic blue with pink halo (*C. tropicalis*), mauve (*C. glabrata*), pink and fuzzy (*C. krusei*), and white (other *Candida* species). *Candida* was passaged on Sabouraud dextrose agar (SDA, Basebio, Hangzhou, China) with 50 μg/mL chloramphenicol (Solarbio, Beijing, China) medium.

**Table 1. Primers sequences and amplification conditions.**

| Primers | Primer sequence (5'-3') | PCR Procedure | Reference |
|---|---|---|---|
| *Hp 16S rDNA* F | GCAATCAGCGTCAGTAATGTTC | 94˚C 45 s, 57˚C 1 min, 72˚C 1 min (33 cycles) | [25] |
| *Hp 16S rDNA* R | GCTAAGAGATCAGCCTATGTCC | | |
| *CagA* F | ATGACTAACGAAACTATTGATCAAACA | 94˚C 30 s, 60˚C 30 s, 72˚C 30 s (35 cycles) | This study |
| *CagA* R | CTGCAAAAGATTGTTTGGCAGA | | |
| *ureA* F | GCCAATGGTAAATTAGTT | 94˚C 1 min, 45˚C 1 min, 72˚C 1 min (40 cycles) | [26] |
| *ureA* R | CTCCTTAATTGTTTTTAC | | |

### Detection of *H. pylori*-specific genes in *Candida* cells

Whole genomic DNA from *Candida* and control isolates was extracted using the UltraClean[®] Microbial DNA Isolation Kit (Qiagen, Germantown, USA). The negative control was *C. albicans* strain ATCC 10231; the positive control was *H. pylori* strain 26695, and the blank control was sterile deionised water. *H. pylori*-specific *16S rDNA*, *cagA* and *ureA* were PCR-amplified in a 25 μL reaction volume containing 1 μL of forward primer (10 μM, Sangon Biotech Co., Ltd., Shanghai, China), 1 μL of reverse primer (10 μM, Sangon Biotech Co., Ltd.), 2 μL of genomic DNA, 12.5 μL of 2× Taq PCR master mix (Jiangsu CWBiotech Science and Technology Co., Ltd., Jiangsu, China), and sterile deionised water (8.5 μL). Primers' sequences and thermocycling conditions are described in Table 1.

Amplicons of *H. pylori*-specific *16S rDNA*, *cagA* and *ureA* were sequenced at Sangon Biotech and aligned with GenBank sequences using the Basic Local Alignment Search Tool (BLAST) (http://www.ncbi.nlm.nih.gov/BLAST/).

**Alignment and phylogenetic analysis.** The evolutionary history is inferred using the Neighbor-Joining method [27]. Alignment of multiple sequences was carried out using ClustalW with a penalty of 15 for gap opening and 6.66 for gap extension. Based on 1000 replicates, a bootstrap consensus tree was inferred to represent the evolutionary history of the taxa. Collapsed branches correspond to partitions that have been replicated less than 50% of the time in bootstrap replications. Next to each branch are the percentages of replicate trees in which the associated taxa clustered together in the bootstrap test (1000 replicates) [28]. Maximum Composite Likelihood was used to determine evolutionary distances and they are expressed in substitutions per site. Using MEGA X, evolutionary analyses were performed on all sequence pairs with ambiguous positions removed (pairwise deletion option).

### Subculturing of *H. pylori*-positive *Candida* and detection of *H. pylori*-specific genes from *Candida*

*Helicobacter pylori*-positive *Candida* were passaged on SDA with chloramphenicol 10 times. To eliminate any possible bacterial contamination, yeast isolates were subcultured on SDA with chloramphenicol ten times. Whole genomic DNA was extracted from the 4th and 10th subcultures, and the *H. pylori*-specific genes were PCR-amplified. Amplicons were sequenced at Sangon Biotech, and genomic alignment was performed using BLAST.

### Microscopic observation of BLBs in *Candida*

Six colonies from the primary culture were randomly selected and placed on slides containing 10 μL of 0.9% saline solution. Coverslips were placed on the samples to search for BLBs under an optical microscope (Nikon, Shanghai, China) equipped with a 100× oil immersion objective lens and camera [29]. *Candida albicans* strain ATCC10231 was used as a negative control.

Furthermore, *Candida* cells were stained using the LIVE/DEAD BacLight Bacterial Viability Kit L7012 (Thermo Fisher, Waltham, MA, USA), which uses a mixture of the SYTO 9 green-fluorescent nucleic acid stain and propidium iodide, a red-fluorescent nucleic acid stain. These stains differ in their spectral characteristics and in their ability to penetrate healthy bacterial cells. When used alone, the SYTO 9 stain generally labels all bacteria (with intact and damaged membranes) in a population. In contrast, propidium iodide penetrates only bacteria with damaged membranes, revealing the bacterial nature of BLBs and their viability. Briefly, fresh *Candida* cultures were suspended in a sterile saline solution, and the turbidity was adjusted to 0.5 McFarland standard. A 0.5 mL of each *Candida* suspension was mixed with 1.5 μL of fluorescent stain containing equal volumes of SYTO 9 and propidium iodide. After a quick vortex and incubation at 25°C in the dark for 15 min, 5 μL of each *Candida* suspension was placed on a glass slide and examined using the 100× lens of a fluorescent microscope equipped with an integrated camera (Nikon, Shanghai, China).

We selected one *H. pylori* 16S rDNA positive vaginal *Candida* strain, one *H. pylori* 16S rDNA positive gastric *Candida* strain, and *C. albicans* ATCC10231 to observe *Candida* cells using transmission electron microscopy (TEM). The fresh *Candida* cultures were collected in 1.5 mL Eppendorf tubes (Eppendorf GmbH, Hamburg, Germany), centrifuged at $10,000 \times g$ for 1 min, and washed thrice with 1 mL phosphate-buffered saline (PBS). The precipitate was resuspended in 2.5% (v/v) glutaraldehyde fixative at 4°C for 24 h and subjected to TEM analysis by Shiyanjia Laboratory (http://www.shiyanjia.com).

## Detection of intracellular *H. pylori* antigen by immunofluorescence

This assay was done using Fluorescein isothiocyanate (FITC)-labeled polyclonal anti-*H. pylori* IgG antibodies at a concentration of 5 mg/mL (Thermo Fisher, Cat # PA1-73161). Each *Candida* isolate was independently cultured in 10 mL of Brain Heart Infusion (BHI) medium (OXOID, Basingstoke, United Kingdom) at 37°C shaking at 120 r/min for 24 h, then *Candida* was treated with 32 μg/mL amphotericin B at 37°C shaking at 120 r/min for 24 h [30]. Amphotericin B increases the permeability of yeast cell walls, which then allows antibodies to enter yeast cells. The culture was then centrifuged for 1 min at 10000 r/min and washed three times with 1 mL of PBS (pH 7.2), the precipitation was resuspended in PBS solution and the turbidity was adjusted to 2 McFarland standard. A 100 μL of each *Candida* suspension was mixed with 1 μL of FITC- labeled anti-*H. pylori* IgG antibodies, and were incubated for 1 h at 25°C in darkness. A 5 μL volume of each suspension was spotted onto a glass slide and observed by fluorescence microscopy. Fresh cultures of *H. pylori* 26695 strain and *C. albicans* ATCC 10231 were used as positive and negative controls, respectively.

## Detection of urease activity in *H. pylori* internalized by *Candida*

To demonstrate intracellular of *H. pylori* release from the *Candida* cells, *H. pylori* 16S rDNA positive *Candida* was inoculated into SDA agar medium (pH 7.2) containing 40% urea (Hai bo, Qingdao, China) and 0.01 g/L phenol red at 37°C for 3–5 days under aerobic conditions. In the presence of *H. pylori*, bacterial urease catalyzes the conversion of urea to ammonia and carbon dioxide, which can be detected by the characteristic red color change in solution [31]. By observing the medium, changing the agar color from yellow to pink is positive for urease and negative for no pink. The *C. albicans* ATCC 10231 was used as negative controls under aerobic conditions (*C. albicans* ATCC 10231 isolated from Man with bronchomycosis). The *H. pylori* 26695 strain was used as a positive control under microaerobic conditions.

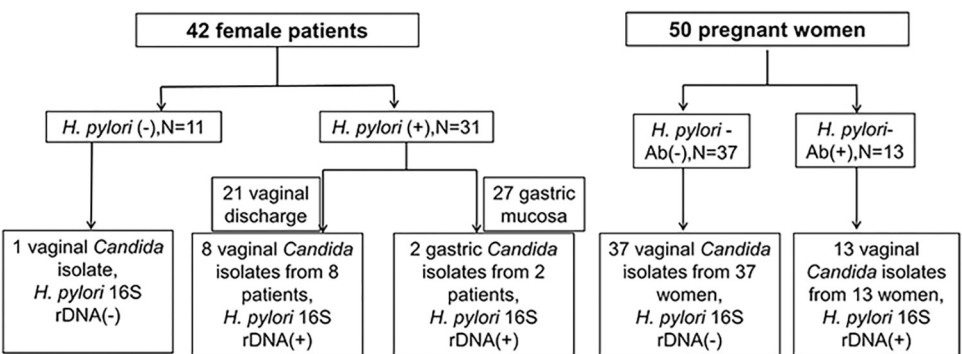

**Fig 1. Consistency of diagnosis across *H. pylori* infection and *H. pylori* within *Candida* cells.** N, number of people*; H. pylori*-Ab (-), *H. pylori* antibody-negative; *H. pylori*-Ab (+), *H. pylori* antibody-positive.

## Results

### *Helicobacter pylori* infection

Of the 50 pregnant women, 13 tested positive for *H. pylori* IgG antibodies. Among the 42 female patients, 31 were $^{14}$C-urea breath test (UBT)-positive and 11 were UBT-negative. Thus, 13 pregnant women and 31 patients with gastric disease were *H. pylori*-positive (Fig 1 and S1 File).

### *Candida* isolation, identification and subcultures

One *Candida* strain was isolated from each vaginal discharge sample to yield 50 isolates. Based on the color of colonies on CHROMagar *Candida* medium, we obtained 45 isolates of *C. albicans*, two isolates of *C. glabrata*, two isolates of *C. tropicalis*, and one isolate of the other *Candida* spp.

Nine vaginal *Candida* isolates and two gastric *Candida* isolates were obtained from female patients with gastric disease, including four *C. albicans*, four *C. cruise*, one *C. glabrata*, and two isolates of the other *Candida* spp. (Fig 1 and Table 2). The *Candida* colonies were sub-cultured on SDA medium for more than ten generations to remove extracellular bacterial contamination.

### Intracellular presence of *H. pylori*-specific genes within *Candida* subcultures

Thirteen vaginal *Candida* isolates from pregnant women, who were positive for *H. pylori* IgG antibody, were positive for intracellular *H. pylori* specific 16S rDNA (Figs 2A and S1) and *ureA* (Fig 2B and S1), and five were also positive for the *cagA* (Figs 3 and S1). In addition, 13 *H. pylori* 16S rDNA positive *Candida* isolates were isolated from pregnant women (Fig 1).

**Table 2. Number of isolated *Candida* from vaginal discharge and gastric mucosa, along with the frequency of 16S rDNA and *cagA* of intracellular *H. pylori* in *Candida*.**

| Participants | Isolated *Candida* | | *Hp* 16S rDNA (+) | | *Hp cagA* (+) | | *Hp urea* (+) | |
|---|---|---|---|---|---|---|---|---|
| | **Vaginal** | **Gastric** | **Vaginal** | **Gastric** | **Vaginal** | **Gastric** | **Vaginal** | **Gastric** |
| **Pregnant women** | 50/50 (100.0%) | NS | 13/50 (26.0%) | NS | 5/13 (38.5%) | NS | 13/13 (100.0%) | NS |
| **Patients with GD** | 9/22 (40.9%) | 2/27 (7.4%) | 8/9 (88.9%) | 2/2 (100.0%) | 5/8 (62.5%) | 2/2 (100.0%) | 6/8(75.0%) | 2/2 (100.0%) |
| **Total** | 59/72 (81.9%) | 2/27 (7.4%) | 21/59 (35.6%) | 2/2 (100.0%) | 10/21 (47.6%) | 2/2 (100.0%) | 19/21(90.5%) | 2/2 (100.0%) |

Numbers in the tables are the numbers of cases; *Hp*, *H. pylori*; GD, gastric disease; NS, not present in any samples; +, positive.

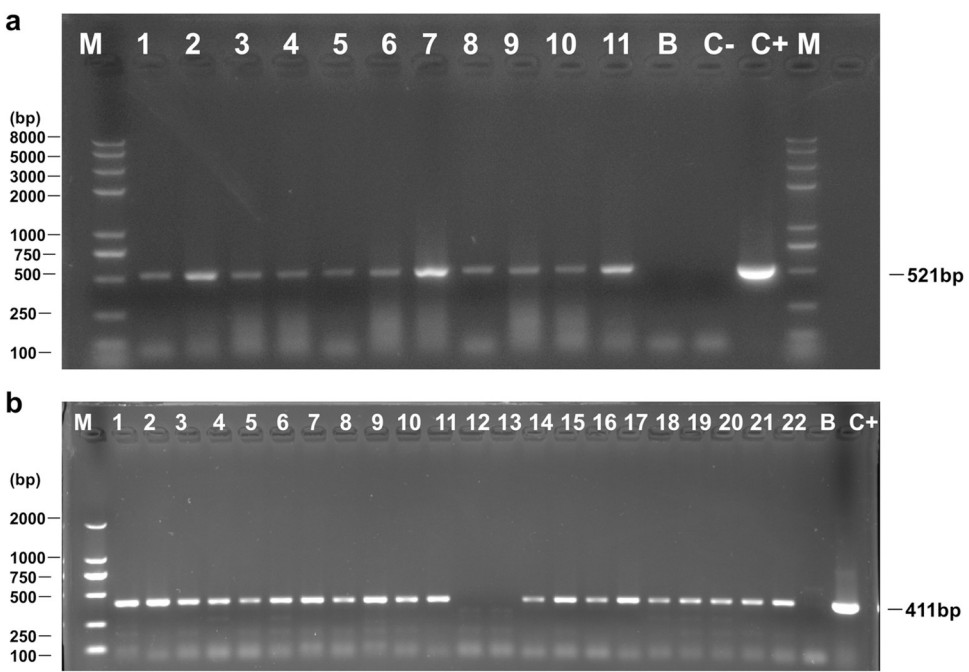

**Fig 2. Agarose gel electrophoresis to visualize PCR-amplified *H. pylori 16S rDNA* and *ureA* from *Candida*.** Lane M: Molecular weight marker. Lane B: Blank control (sterile deionized water). Lane C-: Negative control, DNA extracted from *C. albicans* strain ATCC 10231. Lane C+: Positive control, DNA extracted from *H. pylori* strain 26695. **(a)** Lanes 1–5: DNA extracted from *Candida* harboring BLBs, obtained from vaginal discharge of pregnant women. Lanes 6–8: DNA extracted from *Candida* harboring BLBs, obtained from vaginal discharge of patients with gastric disease. Lanes 10–11: DNA extracted from *Candida* harboring BLBs, obtained from gastric mucosa of patients with gastric disease. **(b)** Lanes 1–11 and Lanes 14–22: DNA extracted from *H. pylori* 16S rDNA positive *Candida*. Lanes 12 and 13: DNA extracted from *H. pylori* 16S rDNA negative *Candida* (S1 Fig).

Eight vaginal *Candida* and two gastric *Candida* isolates from female patients with gastric disease, who were positive for UBT, were positive for intracellular *H. pylori* specific 16S rDNA (Fig 2A), five vaginal *Candida* and two gastric *Candida* were also positive for the *ureA* and *cagA* (Figs 2B and 3). Only one vaginal *Candida* isolate from patients who were negative for UBT was negative for *H. pylori*-specific 16S rDNA and *ureA*.

These PCR positive *Candida* isolates were sub-cultured for 10 generations, 16S rDNA and *cagA* genes of *H. pylori* were still positive for the tenth subcultures. Sequences of these amplicons showed > 99.8% identity with *H. pylori* sequences from GenBank. Some sequences of *H. pylori* 16S rDNA (accession no. ON545841, ON545842 and ON631242) and *cagA* had been submitted to GenBank (accession no. OM779116, OM779117 and OM812999).

### Phylogenetic analyses

The *H. pylori* 16S rDNA sequences obtained from PCR assays were analyzed to determine the evolutionary history of *H. pylori* species within *Candida* cells (Fig 4). Based on BLAST results, the analyses included GenBank reference sequences. These sequences were confirmed to be those of *Helicobacter pylori*.

### Microscopic observations reveal the presence of live *H. pylori* in *Candida* cells

Optical microscopy observations revealed BLBs in 13 of 50 vaginal *Candida* isolates from pregnant women. We also found BLBs in 10 of 11 *Candida* isolates from patients with gastric disease, including eight vaginal isolates and two gastric mucosa isolates (Table 2 and Fig 5). No

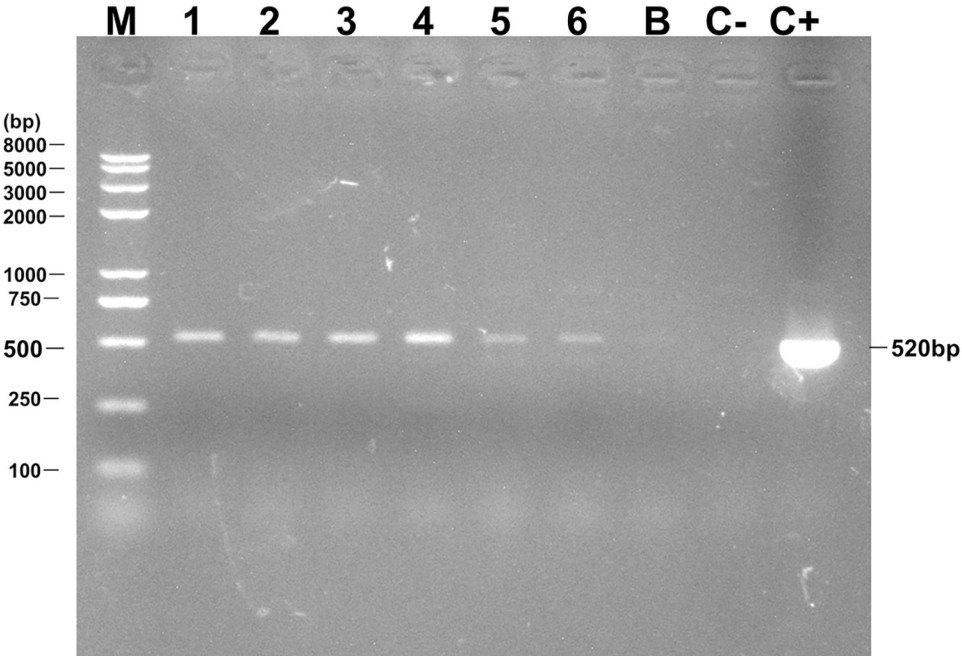

**Fig 3. Agarose gel electrophoresis to visualize PCR-amplified *H. pylori cagA* from *Candida* in patients with gastric disease.** Lane M: Molecular weight marker. Lanes 1–2: DNA extracted from *Candida* isolates in vaginal discharge of pregnant women. Lanes 3–4: DNA extracted from *Candida* isolates in vaginal discharge of female patients with gastric disease. Lanes 5–6: DNA extracted from *Candida* isolates in gastric mucosa of patients with gastric disease. Lane B: Blank control (sterile deionized water). Lane C-: Negative control, DNA extracted from *C. albicans* strain ATCC 10231. Lane C+: Positive control, DNA extracted from *H. pylori* strain 26695 (S1 Fig).

BLBs in *C. albicans* ATCC 10231. At the same time, BLBs were observed to move rapidly within the *Candida* vacuole (S1 Video).

Fluorescence microscopy, using the BacLight Bacterial Viability kit, showed the BLBs within *Candida* cells were viable, as indicated by the green fluorescence (Fig 6). Intracellular *H. pylori* within *Candida* was observed by TEM (Fig 7).

## Direct immunofluorescence assay confirmed the presence of *H. pylori* antigen in *Candida*

The FITC-labeled anti-*H. pylori* IgG antibodies reacted specifically with intracellular *H. pylori* within *Candida* cells as showed by the green fluorescence under fluorescence microscope (Fig 8).

## Detection of urease activity in *H. pylori* internalized by *Candida*

To demonstrate of intracellular *H. pylori* urease activity in *Candida*, *Candida* was inoculated with urea-based SDA agar, which changed the agar color from yellow to pink, indicating urease activity. *H. pylori ureA* positive *Candida* cells in urea-based SDA agar showed urease activity. *C. albicans* ATCC 10231 had no urease activity and *H. pylori* 26695 strain was urease positive.

## Consistency between *H. pylori* infection, BLBs and intracellular *H. pylori* in *Candida*

Thirteen *Candida* isolates were isolated from pregnant women whose *H. pylori* IgG antibody was positive and *H. pylori* 16S rDNA was positive in all *Candida* isolates that harboured BLBs

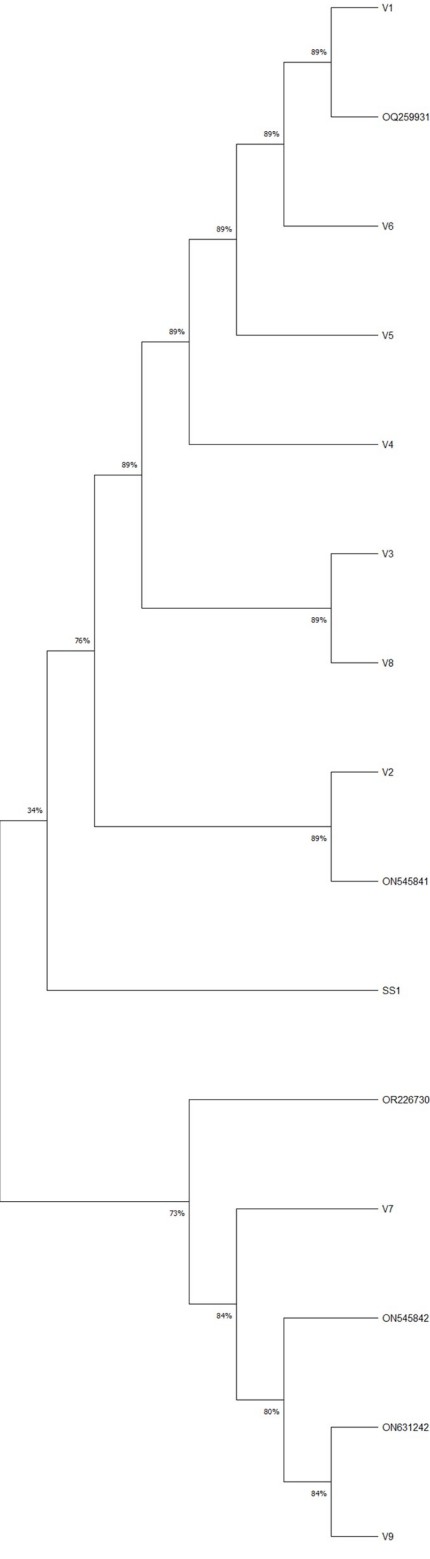

**Fig 4. . An analysis of the *H. pylori* 16S rRNA gene sequences produced a neighbor-joining tree.** This phylogenetic tree was constructed using the GenBank reference sequence (*H. pylori* SS1) and *H. pylori* 16S rRNA sequences obtained from *Candida*. SS1 is a standard strain of *Helicobacter pylori*. V1-V9 and OR226730 represent sequences of *H. pylori* 16S rRNA genes from vaginal *Candida*. ON545841, ON545842 and OQ259931 represent sequences of *H. pylori* 16S rRNA genes from gastric *Candida*.

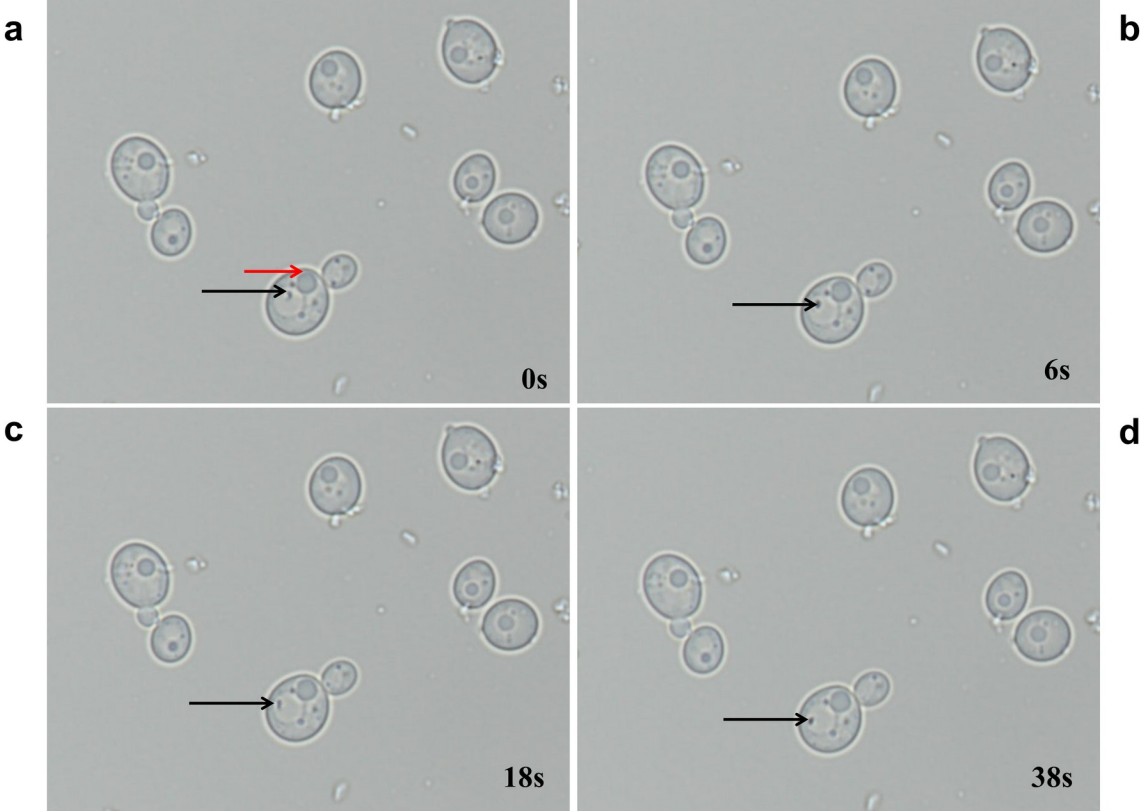

**Fig 5. *Candida* isolated from clinical specimens, under a 100× light microscope.** Photographs taken at 0, 6, 18, and 38 s (a, b, c and d, respectively) show fast-moving BLBs inside vacuoles. Red arrow indicates the *Candida* nucleus, and black arrows indicate BLBs. Refer to the attached video of fast-moving BLBs within vacuoles (S1 Video).

observed by optical microscopy. Interestingly, *H. pylori* 16S rDNA and BLBs were negative in all vaginal *Candida* isolates isolated from pregnant women whose *H. pylori* antibodies were negative (Fig 1). Furthermore, among *H. pylori*-negative patients, only one vaginal *Candida* isolate lacked BLBs and was negative for *H. pylori* 16S rDNA and *ureA* (Fig 1). These results indicate that *H. pylori* 16S rDNA was present in *Candida* isolates with BLBs. Individuals who provided such specimens were therefore infected with *H. pylori*.

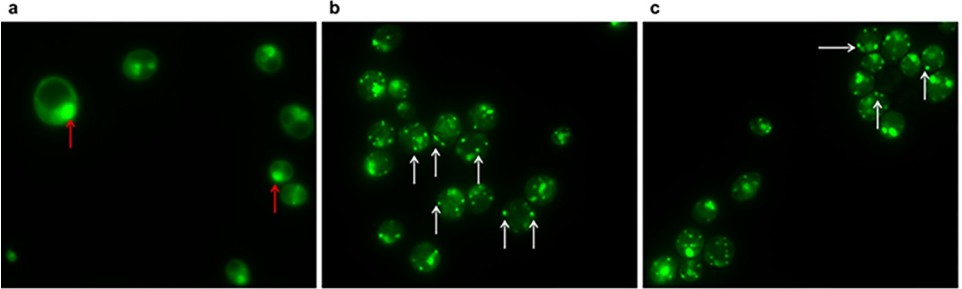

**Fig 6. Fluorescence micrograph of *Candida* stained using a LIVE/DEAD-BacLight kit.** The SYTO 9 stain generally labels all bacteria (with intact and damaged membranes) in a population. In contrast, propidium iodide penetrates only bacteria with damaged membranes. (**a**) *C. albicans* ATCC10231 strain (negative control). (**b**) The live and green BLBs are demonstrated in the *candida* (white arrow), and (**c**) dead *Candida* (magnification × 1000).

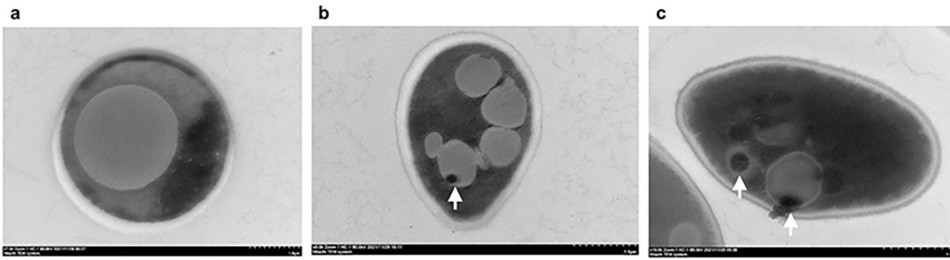

**Fig 7. Transmission electron microscopy (TEM) images of *Candida*. (a)** *C. albicans* ATCC 10231(absence of high electron density body). **(b)** High electron density body (white arrow) in vaginal *Candida* cells. **(c)** High electron density bodies (white arrow) in gastric *Candida* cells. Scale bars = 1.0 μm.

## Discussion

Many studies have reported that *H. pylori* infection exhibits household clustering, with the most likely routes of transmission being oral-oral, fecal-oral, and gastro-oral [8,18,32,33]. High-risk factors for intrafamily transmission include shared towels, mothers feeding pre-chewed food to children, and history of family members with gastrointestinal disease [15]. Despite improvements to social economy, housing, dietary hygiene, and feeding habits within the last 20 years, *H. pylori* infection remains prevalent in children and even infants. From 2009 to 2011 in China, the overall *H. pylori* infection rate in newborns was 0.6%, with some variation across major cities, e.g., 0.3% in Beijing, 3.8% in Chengdu [15]. As newborns rarely have the opportunity to eat contaminated foods, this reflects that their low likelihood of being infected with *H. pylori* through the fecal-oral route. In addition, *H. pylori* is an oxygen-sensitive bacterium that dies soon after leaving the human stomach [34]. Although in most families, mothers assume the responsibility of taking care of children, children and fathers also have close contact. However, in this study, in nine families where only fathers are positive, children are not infected with *H. pylori* [35]. Therefore, the infection route to these newborns' oral cavity is probably directly from their mothers during birth.

Siavoshi et al. reported that vaginal *Candida* may be a host of *H. pylori*, sheltering the bacteria even outside the human body [24]. Subsequently, Sanchez-Alonzo et al. showed that 50% of isolated vaginal *Candida* were positive for *H. pylori* 16S rDNA, and 32% of such samples

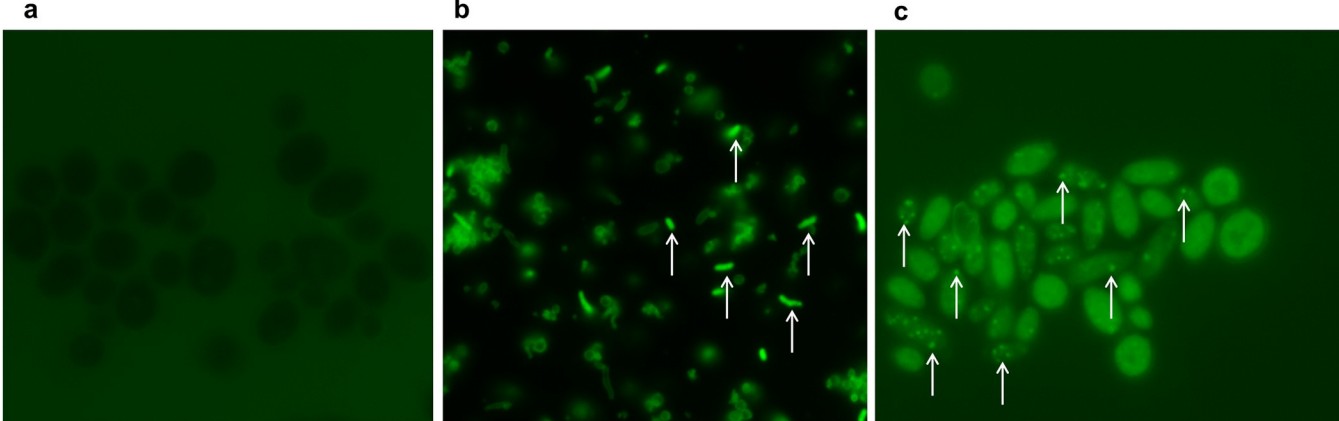

**Fig 8. Immunolabeling of *H. pylori*.** Immunofluorescence micrographs showing: (**a**) absence of fluorescence in the *C. albicans* ATCC 10231 (negative control); (**b**) presence of fluorescence emitted by *H. pylori* 26695 strain (positive control); (**c**) presence of fluorescence emitted by *H. pylori* (white arrows) inside *Candida* (magnification × 1000).

were also positive for *cagA* [29]. CagA is a virulence factor of *H. pylori* infection and is associated with *H. pylori* colonization [36]. Tohidpour reported that CagA wass the first bacterial oncolytic protein to rank the *H. pylori*-mediated adenocarcinoma as the second deadliest cancer type worldwide [37]. These findings corroborate our results. Few reports are available on the relationship between female vaginal *Candida* and *H. pylori* or the vertical transmission of *H. pylori* through the birth canal. Thus, more research is necessary to confirm the intracellular presence of *H. pylori* in *Candida* and its potential transmission from mothers to newborns.

We found that 26% of vaginal *Candida* in pregnant women contained *H. pylori*-specific nucleic acids, consistent with the *H. pylori* IgG positive rates (13/50) in pregnant women. Furthermore, *H. pylori*-specific nucleic acids and BLBs could still be detected even after 10 passages of vaginal *Candida* colonies. Sequences of *H. pylori* 16S rDNA alignments showed > 99.8% identity with *H. pylori* sequences from GenBank. Sequence alignments showed that the base of the *H. pylori* 16S rDNA and *cagA* fragments amplified from different *Candida* isolates had some differences, and no positive amplification from *C. albicans* control isolates, so sample or laboratory contamination with *H. pylori* can be excluded.

As with vaginal *Candida*, *H. pylori*-specific nucleic acids (16S rDNA and *cagA*) were present in *Candida* from the gastric mucosa of *H. pylori*-positive patients, even after ten passages. Researchers have reported that *H. pylori*-specific proteins and *H. pylori* could be detected in gastric yeast by immunofluorescence [38,39]. Amphotericin B increases the permeability of yeast cell walls, which then allows antibodies into yeast cells [30]. In this study, FITC-conjugated anti-*H. pylori* IgG antibodies could be recruited for localization of *H. pylori* inside both vaginal *Candida* and gastric *Candida* cells treated by amphotericin B using immunofluorescence. Therefore, through *Candida*, *H. pylori* could become a facultative intracellular bacterium in the gastrointestinal tract. Sequestration inside yeast vacuoles appears to enhance *H. pylori* survival in gastric acid [40], and an *in vitro* study co-incubating *H. pylori* and *C. albicans* revealed that *H. pylori* enters *C. albicans* vacuoles under unfavorable pH conditions [41]. Thus, *H. pylori* within *Candida* may travel from the stomach to the intestinal tract, and then to the vagina, given its closely proximity to the anus [42]. Several studies have confirmed that *H. pylori* or *H. pylori*-positive *Candida* can enter and colonize the vagina [24,43,44], again pointing to the possibility of *H. pylori*-positive *Candida* being transmitted to newborns during natural delivery.

We detected urease activity in *H. pylori* internalized by *Candida*, which turned the urea-based SDA pink, demonstrating the release of intracellular *H. pylori* urease from *Candida* cells, as well as the presence of the *H. pylori ureA* gene. Heydari et al. reported that *Candida* may release free *H. pylori* as a vesicle-encased or free bacterium [45]. While the elimination of numerous extracellular *H. pylori* isolates can relieve symptoms, *Candida* reservoirs of *H. pylori* may cause chronic infections, recurrence, or even carcinogenesis. The fact that *H. pylori* was found in *Candida* isolated from the stomach after several passages indicates that *H. pylori* remains in *Candida* progeny even after the *Candida* cells bud.

Additionally, under light microscopy, we observed rapid movement of bacteria within *Candida* vacuoles, similarly to previous reports on oral *Candida* [46]. This movement increases the probability that *H. pylori* can proliferate within *Candida* cells and use them as a vector for transmission. Although intracellular *H. pylori* in *Candida* cannot be isolated and cultured, Heydari et al. reported that the released *H. pylori* from *Candida* were detected by immuno-magnetic separation [44].

As early as 2014, Siavoshi et al. considered vacuoles of *Candida* cell as a specialized niche for *H. pylori* [47]. However, this conclusion has not received much attention, and even was opposed by Alipour and Gaeini [48], who thought that the size of *Candida* cells was not large enough to accommodate several *H. pylori* cells. Additionally, the *H. pylori* isolates cannot be

isolated from *Candida*, which would violate Koch's postulates [48]. Then, Siavoshi et al. quickly made a reasonable response, insisted on the published research conclusions, and considered that the traditional Koch's postulates are not applicable to the non-culturable *H. pylori* in *Candida* vesicles [49]. The size of a *Candida* cell is ~3–8 μm [50], the bacterial cell length of *H. pylori* is ~2–4 μm and its width is ~0.5–1 μm [51]. Although it is difficult for *Candida* to accommodate several *H. pylori* cells, under the exposure of antibiotics, *H. pylori* may lose its cell wall and become a coccoid form, which can be internalized into *Candida* cells. Therefore, under unfavorable conditions such as exposure to antibiotics and an aerobic environment, the vacuoles of *Candida* cells provided a special habitat for *H. pylori* and made it a shelter for *H. pylori* [39,47]. *Candida albicans* is an opportunistic fungal pathogen that can colonize host niches at varying pH [52]. In our study, *Candida* species carrying *H. pylori* showed detectable urease activity. It suggests a symbiotic relationship between *H. pylori* and *Candida*, allowing the yeast to survive in acidic environments.

Overall, our work provides a clue for further in-depth study on the interaction between *H. pylori* and *C. albicans*, routes of *H. pylori* transmission, and pathogenicity.

## Conclusion

In conclusion, our results showed that *H. pylori* and its virulence-related genes were present in *Candida* from vaginal discharge of pregnant women. We also found *H. pylori* in vaginal *Candida* and gastric *Candida* from female patients with gastric disease, with *H. pylori* 16S rDNA was still detectable after 10 subcultures of *Candida*. As a result, *Candida* may act as a reservoir for *H. pylori* in women's gastrointestinal tract, allowing the bacteria to migrate through the intestines to the vagina and eventually infect newborns during delivery.

## Supporting information

**S1 Fig.**
(PDF)

**S1 File.**
(XLSX)

**S1 Video.**
(AVI)

## Acknowledgments

We thank the Department of Obstetrics in Jinyang Affiliated Hospital of Guizhou Medical University and the Department of Gastroenterology in the People's Hospital of Qiannan Prefecture (Guizhou Province) for collecting specimens used in this study. We also acknowledge our colleagues for their valuable comments on this paper. We would also like to thank Editage [www.editage.cn] for their help with English-language editing.

## Author Contributions

**Conceptualization:** Tingxiu Yang, Guzhen Cui, Zhenghong Chen.

**Data curation:** Tingxiu Yang, Zhenghong Chen.

**Formal analysis:** Tingxiu Yang, Zhenghong Chen.

**Funding acquisition:** Zhenghong Chen.

**Investigation:** Zhenghong Chen.

**Methodology:** Tingxiu Yang, Jia Li, Yuanyuan Zhang, Zhaohui Deng, Jianchao Sun, Xiaojuan Wu, Dengxiong Hua.

**Project administration:** Tingxiu Yang, Zhenghong Chen.

**Resources:** Jia Li, Yuanyuan Zhang, Zhaohui Deng, Jun Yuan.

**Software:** Tingxiu Yang, Guzhen Cui.

**Supervision:** Zhenghong Chen.

**Validation:** Tingxiu Yang, Zhenghong Chen.

**Visualization:** Tingxiu Yang, Xiaojuan Wu, Song Xiang.

**Writing – original draft:** Tingxiu Yang, Jia Li.

**Writing – review & editing:** Tingxiu Yang, Jia Li, Yuanyuan Zhang, Guzhen Cui, Jianchao Sun, Xiaojuan Wu, Dengxiong Hua, Zhenghong Chen.

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
