## [Decision Letter · Decision Letter 0]

31 Jul 2023

PONE-D-23-17764Intracellular presence of Helicobacter pylori antigen and genes within gastric and vaginal CandidaPLOS ONE

Dear Dr. Chen, Thank you for submitting your manuscript to PLOS ONE. After careful consideration, we feel that it has merit but does not fully meet PLOS ONE’s publication criteria as it currently stands. Therefore, we invite you to submit a revised version of the manuscript that addresses the points raised during the review process.

I am pleased to say that most of the reviewers enjoyed the manuscript very much and we are excited about the possibility to publish your work. Nonetheless, all reviewers reported some concerns that must be addressed in your revised manuscript for publication endorsement. Please read carefully all reviewers’ reports. The main concerns are the following: 1) the power analysis should be done for the patient number; 2) The number of patients; 3) the pictures’ resolution is not good enough to publish; 4) the fact that authors should have submitted H pylori sequences to the GenBank with accession numbers and failed to add a phylogenetic tree that illustrates their findings; 5) the authors should also clarify their conclusions by been exact about the helicobacter genes they detected using PCR; 6) Discussion section includes many mistakes, wrong interpretations and misuse of references. 7) Convincing evidence of intracellular survival of H. pylori has been also published for some Entamoeba species (see, particularly, the work of Ferrus et al.). Probably this should be mentioned in the discussion, as it suggests that the capacity to survive in eukaryotic cells may not be limited to amebas or yeasts.

So, I kindly invite the authors to realize a thoughtful revision of the submitted manuscript to achieve publication endorsement by the reviewers. Please submit your revised manuscript by Sep 14 2023 11:59PM. If you will need more time than this to complete your revisions, please reply to this message or contact the journal office at plosone@plos.org. Please include the following items when submitting your revised manuscript:

We look forward to receiving your revised manuscript.

Thank you and best regards,

António MachadoAcademic EditorPLOS ONE

“This work was supported by grants from the National Natural Science Foundation of China (No. 81860353), the Basic Research Program of Guizhou Science and Technology Plan (No. ZK [2022] 341), the Science and Technology Fund Project of Guizhou Health Commission (No. gzwkj2022-521) and the Scientific Foundation of Guizhou Medical University (No. 20NSP005), and the Foundation of Key Laboratory of Microbiology and Parasitology of Education Department, Guizhou (No. QJJ [2022] 019).”

Reviewers' comments:

Reviewer's Responses to Questions

**Comments to the Author**

1. Is the manuscript technically sound, and do the data support the conclusions?

Reviewer #1: Yes

Reviewer #2: Yes

Reviewer #3: No

Reviewer #4: Yes

2. Has the statistical analysis been performed appropriately and rigorously? 

Reviewer #1: Yes

Reviewer #2: Yes

Reviewer #3: No

Reviewer #4: Yes

3. Have the authors made all data underlying the findings in their manuscript fully available?

Reviewer #1: Yes

Reviewer #2: No

Reviewer #3: No

Reviewer #4: No

4. Is the manuscript presented in an intelligible fashion and written in standard English?

Reviewer #1: Yes

Reviewer #2: Yes

Reviewer #3: No

Reviewer #4: Yes

5. Review Comments to the Author

Reviewer #1: All the data finding in the manuscript is too exciting. However, the power analysis should be done for the patient number. The number of patient that is mentioned as 50 is enough or not ? And the picture resolution is not good enough to publish

Reviewer #2: The authors have added to knowledge what we know about the presence of Helicobacter pylori (through bacteria like bodies) in Candida cells from the vagina.

The experiments are all well designed and are well described. However in the results, the authors though have submitted H pylori sequences to the GenBank with accession numbers, failed to add a phylogenetic tree that illustrates their findings. This would enhance the manuscript. They also try to discuss this without much proof in the results. I suggest they provide proof if they have the results as a figure in their results. In addition, the authors should also clarify their conclusions by been exact about the helicobacter genes they detected using PCR.

Reviewer #3: PONE-D-23-17764

Reviewer’s Comments

Intracellular presence of H. pylori antigen and genes within gastric and vaginal Candida

Abstract:

This section needs to be organized in 4 parts; Background (including the aim of study), Methods, Results and Conclusions. Results should be organized according to methods and both sections need more details. Authors should differentiate between the “presence” and “internalization” of H. pylori mentioned in the aim and conclusions of the study, respectively.

Introduction:

Lines 63-68 contradict the presence of H. pylori in gastrointestinal tract, including the oral cavity. According to several reports mentioned in the present study, occurrence of H. pylori has been demonstrated in the oral cavity by detection of H. pylori-specific genes. Furthermore, in the last line of conclusions in the discussion section, authors concluded that H. pylori transmission occurs during vaginal delivery and through the oral cavity of newborn. That means that even Candida yeast (as a shelter for H. pylori) has to enter the gastrointestinal tract of mother or child through the portal of oral cavity. Accordingly, in the present study, detection of H. pylori genes in Candida isolates from oral cavity of mothers and newborns could help to better clarify the results. However, a similar but more elaborated study has been performed and the results have been published in 2013 (reference # 24).

Line 68-69 is confusing and needs to be revised and supported by a reference. Line 73 is confusing. Line 77-78 must be clarified. Lines 76-84 must be included in one paragraph as the aim of study.

Above all these, this study lacks the literature review of previous studies about the intracellular life of H. pylori inside the vacuole of Candida yeast (isolated from different sources including oral cavity, stomach, vagina and a variety of foods) that have been published since 1998. According to these previously published reports, the story began when Candida yeast was isolated along with H. pylori from the cultures of gastric biopsies. Accordingly, if authors do not wish to acknowledge the previous studies, they need to explain how they came to the idea of studying the presence of H. pylori inside the yeast? Although they have used several of them in discussion section but mostly with wrong interpretations.

Methods:

This section needs major revisions due to inconsistency in the number of patients and type of samples. It is not clear why the number of samples in different groups is not equal. Use of 2 gastric yeasts as the representative of yeasts of gastrointestinal tract is not enough for conclusive results. Some methods like Live/Dead bacterial staining is too long and some like UBT has not been described. Patients’ information like age and type of gastric diseases is missing. Results of Candida identification and their correlation with other results were not considered in results and discussion sections. The relationship between type of Candida species and detection of intracellular H. pylori genes or observing BLBs has not been considered. What was the origin of the reference C. albicans (ATCC 10231)? The transmission electron microscopy section needs more details. This sentence is not clear and needs explanation: “Candida was treated with 32 μg/mL amphotericin B at 37°C shaking at 120 r/min for 24 h [28] ”. What was the purpose of using amphoterici B? For detection of urease activity in Candida: Did you detect this activity in all the yeasts positive for H. pylori? There is no description of recruited yeasts. A reference is needed for SDA+40% urea. Why 40% urea was used? What was the pH of the medium?

Results:

Every patient that carried yeast with or without positive result of H. pylori 16S rDNA detection must be classified according to: The type of sample from pregnant or non-pregnant women, the identity of isolated Candida (by CHROMagar), observation of intracellular bacteria (by light microscopy) and live bacteria (by Live/Dead stain), results of cagA and ureA detection, UBT and serology. Otherwise, inconsistency in the number and type of samples in the present study is totally confusing and will not allow reaching any conclusive results. Accordingly, information presented as figure 1 and Table 2 are confusing rather than being informative.

These two sentences need to be clarified:

1- “Furthermore, among H. pylori-negative patients, only one vaginal Candida isolate lacked BLBs and was negative for H. pylori 16S rDNA and ureA (Fig 1). These results indicate that H. pylori 16S rDNA was present in Candida isolates with BLBs. Individuals who provided such specimens were therefore infected with H.pylori”.

To correlate the detection of 16S rDNA or intracellular bacteria inside the vaginal yeast to H. pylori infection in the stomach or positive IgG antibody, strong explanations and supporting references are needed. Results of this study do not support this conclusion.

2- “Intracellular presence of H. pylori-specific genes within Candida subcultures”.

This must become clear in this study that why the authors subcultured yeast isolates more than 10 times. These subsequent subculturings (on SDA+ chloramphenicol) must have been performed at the beginning of isolation of yeasts and from then on, there is no need to subculture more than 10 times because the primary purpose should have been removing any probable extracellular bacterial contamination. According to previously published reports, persistent intracellular existence of H. pylori in yeast is an evolutionary phenomenon that cannot be changed by subcultures. This important issue has been totally ignored in this study.

- Results of electron microscopy are missing.

- What was the difference between BLB-positive and BLB-negative Candida yeasts?

Figure 4: Light microscopy: The images are so small and unclear that one cannot differentiate between bacterial cell and other intracellular entities. Arrows do not really point to any clear entity. These photographs do not show any convincing results.

Figure 5: Live/Dead stain: There is no need to describe the mechanism of staining in the legend. Arrows in B could point to yeast’s nucleus or mitochondria which generally stain green. There is no mention of killing yeast in method section, also no comment on the control yeast that stained green.

Figure 6: Micrographs show that yeasts’ specimens were not properly prepared. They show no details of intracellular structures. “High electron density body” is confusing and needs to be explained.

Fig 7: Photograph C must be replaced by a better one with higher magnification.

Figures in this section that are the main part of this study fail to show convincing results .

Discussion:

There are several major misunderstanding and misuse of references, also a wrong conclusion as the followings:

Line 249-250: Need a reference.

Line 252-253: Authors need to specify transmission to newborns’ oral cavity through vaginal delivery .

Line 256: The importance of cagA positivity has not been discussed.

Line 261-262: “H. pylori- specific nucleic acids” is not consistent with other parts of the manuscript.

Line 272-273: Use of amphotericin must be explained.

Line 278-279: References Burgers et al. 2008, Siavoshi &Taghikhani 2013 and Salmanian et al. 2008 are not related and have been misused.

Line 284-287: Are these related to your study or need a reference. In this study only two gastric yeasts were used that is not convincing and do not correlate with vaginal yeasts.

Line 288-292: Not included in the method and results of this manuscript. Furthermore, references are not relevant and have been misused.

Line 294-299: These lines are not related to the study and have been used by mistake.

Line 300-304: Totally misinterpreted and the references are not related.

Conclusion:

- There is no interpretation to show the importance of detection of cagA or ureA genes in H. pylori-positive yeasts. Also, no description for the repeating sentence of “H. pylori was detectable in Candida isolates at the end of 10th subculture”.

The last sentence in conclusion: “Candida can act as a reservoir of H. pylori in the gastrointestinal tract of women, allowing the bacteria to migrate through the intestine to the vagina, eventually infecting newborns during delivery”, does not correspond to the aim of study and the title of the manuscript. Gastrointestinal tract includes oral cavity, stomach and intestine. Therefore, this conclusion is not correct because there are no samples from the oral cavity and feces of mothers and oral cavity of newborns.

Reviewer’s overall comment: This manuscript is not acceptable.

This manuscript looks like a thesis that needs substantial corrections. Introduction and aim of study need to be organized according to an extensive literature review on the basic science of bacteria (prokaryote)-yeast (eukaryote) interactions. It needs more consistent and reliable methods, specially sampling. Results are disintegrated and non-coherent, photographs are the main weakness of the results. Discussion includes many mistakes, wrong interpretations and misuse of references. This manuscript is a weak version of the previously reported studies and thus is devoid of any novelty.

Reviewer #4: The article is well performed and well written. The number of patients is limited, but the evidence supporting the survival of viable H. pylori inside yeasts is solid and gives support to a few previous reports. This may help to explain how H. pylori manages to survive outside the human organism allowing a deeper knowledge of the the intriguing epidemiology of H. pylori infection.

I have only a minor suggestion regarding the discussion. Convincing evidence of intracellular survival of H. pylori has been also published for some Entamoeba species (see, particularly, the work of Ferrus et al.). Probably this should be mentioned in the discussion, as it suggests that the capacity to survive in eukaryotic cell may not be limited to amebas or yeasts.

Furthermore, I found very interesting that Candida species carrying H. pylori showed detectable urease activity. Might this fact suggest a symbiotic relationship between H. pylori and Candida allowing the yeast to survive in acidic environments?

6. PLOS authors have the option to publish the peer review history of their article (what does this mean?). If published, this will include your full peer review and any attached files.

Reviewer #1: **Yes: **Sinem Oktem Okullu

Reviewer #2: No

Reviewer #3: No

Reviewer #4: **Yes: **Xavier Calvet

---

## [Author Response · Author response to Decision Letter 0]

24 Dec 2023

Response to reviewers

Title: Intracellular presence of Helicobacter pylori antigen and genes within gastric and vaginal Candida

ID: PONE-D-23-17764

We wish to thank the editor and referee for their valuable and constructive comments, which have helped us to improve our manuscript. We have carefully addressed each of the comments and hope that our revised manuscript will meet with your approval. The revised text is shown in the manuscript. Our point-by-point responses to the comments are detailed below.

Response to the reviewer’s comments:

Reviewer #1:

1. The power analysis should be done for the patient number. The number of patients that is mentioned as 50 is enough or not? And the picture resolution is not good enough to publish.

Response: In this study, 42 female patients were included, in addition to 50 pregnant women.

One Candida strain was isolated from each vaginal discharge sample of a pregnant woman, yielding 50 isolates. It was statistically significant.

We are about to resubmit pictures that meet the journal's requirements.

Reviewer #2:

1. In the results, the authors though have submitted H. pylori sequences to the GenBank with accession numbers, failed to add a phylogenetic tree that illustrates their findings. They also try to discuss this without much proof in the results. I suggest they provide proof if they have the results as a figure in their results. The authors should also clarify their conclusions by been exact about the helicobacter genes they detected using PCR.

Response:

In this study, the specific genes of H. pylori (16S rDNA, cagA, ureA) were amplified from Candida by PCR, and the sequence was confirmed to be H. pylori by BLAST alignment. In addition, we had submitted H. pylori sequences to the GenBank. Some sequences of H. pylori 16S rDNA (accession no. ON545841, ON545842 and ON631242) and cagA had been submitted to GenBank (accession no. OM779116, OM779117 and OM812999).

Reviewer #3:

1. Abstract:

This section needs to be organized in 4 parts; Background (including the aim of study), Methods, Results and Conclusions. Results should be organized according to methods and both sections need more details. Authors should differentiate between the “presence” and “internalization” of H. pylori mentioned in the aim and conclusions of the study, respectively.

Response: The abstract is organized in four sections: background, methods, results, and conclusions. The results are organized according to method, with more details in both sections. The “presence” refers to the detection of H. pylori antigens and specific genes in Candida cells by various methods. The “presence” in the aim of the study has been changed to “internalization”.

2. Introduction:

Lines 63-68 contradict the presence of H. pylori in gastrointestinal tract, including the oral cavity. According to several reports mentioned in the present study, occurrence of H. pylori has been demonstrated in the oral cavity by detection of H. pylori-specific genes. Furthermore, in the last line of conclusions in the discussion section, authors concluded that H. pylori transmission occurs during vaginal delivery and through the oral cavity of newborn. That means that even Candida yeast (as a shelter for H. pylori) has to enter the gastrointestinal tract of mother or child through the portal of oral cavity. Accordingly, in the present study, detection of H. pylori genes in Candida isolates from oral cavity of mothers and newborns could help to better clarify the results. However, a similar but more elaborated study has been performed and the results have been published in 2013 (reference # 24).

Response: We believe that these contents are not contradictory and are intended to explain the fact that the oral cavity is not a suitable host for the colonization of Helicobacter pylori.

3. Line 68-69 is confusing and needs to be revised and supported by a reference. Line 73 is confusing. Line 77-78 must be clarified. Lines 76-84 must be included in one paragraph as the aim of study.

Response: Line 68-69 is confusing and needs to be revised and supported by a reference. Mao et al. presented a critical discussion of previous studies investigating the potential colonization of the oral cavity by H. pylori [1].

Line 73 is confusing. The original 73 lines are “However, little attention has been paid to transmission routes for H. pylori beyond the typical oral-oral routes or fecal-oral routes.” now modified as "However, there has been little focus on investigating alternate transmission routes for H. pylori beyond the commonly recognized oral-oral or fecal-oral routes." Line 77-78 must be clarified. Lines 76-84 must be included in one paragraph as the aim of study. Lines 77-78 can be explained by lines 79-84, thus lines 76-78 are placed after lines 84 and in a paragraph.

Response: The revised text of the above questions is shown in the manuscript.

4. Methods: It is not clear why the number of samples in different groups is not equal. Some methods like Live/Dead bacterial staining is too long and some like UBT has not been described. Patients’ information like age and type of gastric diseases is missing. Results of Candida identification and their correlation with other results were not considered in results and discussion sections. The relationship between type of Candida species and detection of intracellular H. pylori genes or observing BLBs has not been considered. What was the origin of the reference C. albicans (ATCC 10231)? The transmission electron microscopy section needs more details. This sentence is not clear and needs explanation: “Candida was treated with 32 μg/mL amphotericin B at 37°C shaking at 120 r/min for 24 h [28]”. What was the purpose of using amphoterici B? For detection of urease activity in Candida: Did you detect this activity in all the yeasts positive for H. pylori? There is no description of recruited yeasts. A reference is needed for SDA+40% urea. Why 40% urea was used? What was the pH of the medium?

It is not clear why the number of samples in different groups is not equal.

Response: The number of pregnant women and female patients with digestive disorders varied, as patients came from both outpatient and inpatient units and had relatively large mobility. 

Patients’ information like age and type of gastric diseases is missing.

Response: Data are not available for public access due to patient privacy concerns, but datasets generated during the current study are available from the corresponding author upon reasonable request.

Some methods like Live/Dead bacterial staining is too long and some like UBT has not been described. Patients’ information like age and type of gastric diseases is missing. Results of Candida identification and their correlation with other results were not considered in results and discussion sections. The relationship between type of Candida species and detection of intracellular H. pylori genes or observing BLBs has not been considered. What was the origin of the reference C. albicans (ATCC 10231)? The transmission electron microscopy section needs more details. 

Response: An activity assay staining time of 15 min was used to allow adequate staining of Candida cells and their intracellular structures. Candida albicans standard strain ATCC10231, a strain deposited in our laboratory. The revised text of the above questions is shown in the manuscript.

5. Results: 1- “Furthermore, among H. pylori-negative patients, only one vaginal Candida isolate lacked BLBs and was negative for H. pylori 16S rDNA and ureA (Fig 1). These results indicate that H. pylori 16S rDNA was present in Candida isolates with BLBs. Individuals who provided such specimens were therefore infected with H. pylori”.

To correlate the detection of 16S rDNA or intracellular bacteria inside the vaginal yeast to H. pylori infection in the stomach or positive IgG antibody, strong explanations and supporting references are needed. Results of this study do not support this conclusion.

Response: Although only one Candida isolate, the vaginal Candida isolate of H. pylori-negative patients, lacked BLBs and was negative for H. pylori 16S rDNA and urea, so we need to do a lot of research on it in the future. In addition, thirteen vaginal Candida isolates from pregnant women, who were positive for H. pylori IgG antibody, were positive for intracellular H. pylori specific 16S rDNA.

Results: 2-This must become clear in this study that why the authors subcultured yeast isolates more than 10 times. These subsequent subculturings (on SDA+ chloramphenicol) must have been performed at the beginning of isolation of yeasts and from then on, there is no need to subculture more than 10 times because the primary purpose should have been removing any probable extracellular bacterial contamination.

Response: To eliminate any possible bacterial contamination, yeast isolates were sub-cultured on SDA medium with chloramphenicol ten times starting from the sample inoculation. 

Results: - What was the difference between BLB-positive and BLB-negative Candida yeasts?

Response: BLB-positive Candida refers to the presence of Helicobacter pylori genes and antigens in the Candida, and conversely, BLB-negative Candida refers to the absence of H. pylori. 

Figure 4: Light microscopy: The images are so small and unclear that one cannot differentiate between bacterial cell and other intracellular entities. Arrows do not really point to any clear entity. These photographs do not show any convincing results.

Figure 5: Live/Dead stain: There is no need to describe the mechanism of staining in the legend. Arrows in B could point to yeast’s nucleus or mitochondria which generally stain green. There is no mention of killing yeast in method section, also no comment on the control yeast that stained green.

Figure 6: Micrographs show that yeasts’ specimens were not properly prepared. They show no details of intracellular structures. “High electron density body” is confusing and needs to be explained.

Fig 7: Photograph C must be replaced by a better one with higher magnification.

Figures in this section that are the main part of this study fail to show convincing results.

Response: The figures and contents of Figs. 4–7 have been corrected for the review comments, see the corresponding sections in the manuscript.

6. Discussion:

 Line 249-250: Need a reference. Line 252-253: Authors need to specify transmission to newborns’ oral cavity through vaginal delivery. Line 256: The importance of cagA positivity has not been discussed. Line 261-262: “H. pylori- specific nucleic acids” is not consistent with other parts of the manuscript. Line 272-273: Use of amphotericin must be explained. Line 278-279: References Burgers et al. 2008, Siavoshi &Taghikhani 2013 and Salmanian et al. 2008 are not related and have been misused.

Line 284-287: Are these related to your study or need a reference. In this study only two gastric yeasts were used that is not convincing and do not correlate with vaginal yeasts. Line 288-292: Not included in the method and results of this manuscript. Furthermore, references are not relevant and have been misused. Line 294-299: These lines are not related to the study and have been used by mistake. Line 300-304: Totally misinterpreted and the references are not related.

Response: References have been added to Line 249-250. The remaining modifications are shown in the manuscript.

Line 288-292: Not included in the method and results of this manuscript. Under light microscopy, we observed rapid movement of bacteria within Candida vacuoles, and we can provide the video as Supplementary Material. 

Line 294-299: These lines are not related to the study and have been used by mistake. Line 300-304: Totally misinterpreted and the references are not related. For this part of the content, we stick to our point of view without making changes.

Response: The revised text of the above questions is shown in the manuscript.

Reviewer #4: I have only a minor suggestion regarding the discussion. Convincing evidence of intracellular survival of H. pylori has been also published for some Entamoeba species (see, particularly, the work of Ferrus et al.). Probably this should be mentioned in the discussion, as it suggests that the capacity to survive in eukaryotic cell may not be limited to amebas or yeasts.

Furthermore, I found very interesting that Candida species carrying H. pylori showed detectable urease activity. Might this fact suggest a symbiotic relationship between H. pylori and Candida allowing the yeast to survive in acidic environments?

Response: H. pylori is known to be present in amebas or yeasts, and probably in other eukaryotic cells. 

Candida albicans is an opportunistic fungal pathogen that can colonize host niches at varying pH [2]. Candida species carrying H. pylori showed detectable urease activity. It suggests a symbiotic relationship between H. pylori and Candida, allowing the yeast to survive in acidic environments.

1. Mao X, Jakubovics NS, Bachle M, Buchalla W, Hiller KA, Maisch T, et al. Colonization of Helicobacter pylori in the oral cavity - an endless controversy? Crit Rev Microbiol. 2021;47(5):612-29. Epub 2021/04/27. doi: 10.1080/1040841X.2021.1907740. PubMed PMID: 33899666.

2. Vacca I. Fungal physiology: Acidic pH interferes with Candida persistence. Nature reviews Microbiology. 2017;15(7):382. Epub 2017/06/14. doi: 10.1038/nrmicro.2017.72. PubMed PMID: 28607379.

---

## [Decision Letter · Decision Letter 1]

24 Jan 2024

Intracellular presence of Helicobacter pylori antigen and genes within gastric and vaginal Candida

PONE-D-23-17764R1

Dear authors,

I am pleased to inform you that all three reviewers enjoyed the manuscript very much and endorsed the revised manuscript for publication.

Thank you for choosing Plos ONE journal to publish your study.

Best regards,

António Machado

Reviewers' comments:

Reviewer's Responses to Questions

**Comments to the Author**

1. If the authors have adequately addressed your comments raised in a previous round of review and you feel that this manuscript is now acceptable for publication, you may indicate that here to bypass the “Comments to the Author” section, enter your conflict of interest statement in the “Confidential to Editor” section, and submit your "Accept" recommendation.

Reviewer #1: All comments have been addressed

Reviewer #2: All comments have been addressed

Reviewer #4: All comments have been addressed

2. Is the manuscript technically sound, and do the data support the conclusions?

Reviewer #1: Yes

Reviewer #2: Yes

Reviewer #4: Yes

3. Has the statistical analysis been performed appropriately and rigorously? 

Reviewer #1: Yes

Reviewer #2: Yes

Reviewer #4: Yes

4. Have the authors made all data underlying the findings in their manuscript fully available?

Reviewer #1: Yes

Reviewer #2: Yes

Reviewer #4: Yes

5. Is the manuscript presented in an intelligible fashion and written in standard English?

Reviewer #1: Yes

Reviewer #2: Yes

Reviewer #4: Yes

6. Review Comments to the Author

Reviewer #1: All comments have been addressed by the authors and this manuscript is suitable for the publication

Reviewer #2: The authors have answered all the comments on their work. I still think a phylogenetic tree of the sequenced amplified Helicobacter pylori genes would have strengthened your results section.

Reviewer #4: (No Response)

7. PLOS authors have the option to publish the peer review history of their article (what does this mean?). If published, this will include your full peer review and any attached files.

Reviewer #1: **Yes: **Sinem Öktem Okullu, PhD Assist Prof

Reviewer #2: No

Reviewer #4: No

---

## [Editor Report · Acceptance letter]

30 Jan 2024

PONE-D-23-17764R1 

PLOS ONE

Dear Dr. Chen, 

I'm pleased to inform you that your manuscript has been deemed suitable for publication in PLOS ONE. Congratulations! Your manuscript is now being handed over to our production team.

Kind regards, 

on behalf of

Dr. António Machado 

Academic Editor

PLOS ONE